Mapping open knowledge institutions: an exploratory analysis of Australian universities

Huang Chun-Kai (Karl) 1 karl.huang@curtin.edu.au
http://orcid.org/0000-0001-8705-1027 Wilson Katie 1
http://orcid.org/0000-0002-0068-716X Neylon Cameron 1 2
http://orcid.org/0000-0001-6813-8362 Ozaygen Alkim 1
http://orcid.org/0000-0001-6551-8140 Montgomery Lucy 1 2
http://orcid.org/0000-0001-8288-5241 Hosking Richard 1 2
1 Centre for Culture and Technology, Curtin University , Bentley, Western Australia , Australia
2 Curtin Institute for Computation, Curtin University , Bentley, Western Australia , Australia
Stern David
Electronic publication date: 2021 May 11
Publication date: 2021
Volume: 9
Electronic Location ID: e11391
Received 2020 Sep 25; Accepted 2021 Apr 12
Copyright: © 2021 Huang et al.
Copyright year: 2021
Copyright holder: Huang et al.
License: This is an open access article distributed under the terms of the Creative Commons Attribution License, which permits unrestricted use, distribution, reproduction and adaptation in any medium and for any purpose provided that it is properly attributed. For attribution, the original author(s), title, publication source (PeerJ) and either DOI or URL of the article must be cited.
License URL: https://creativecommons.org/licenses/by/4.0/

Keywords: Open knowledge institutions, Open access, Diversity, Principal component, Altmetrics, Scientometrics, Universities, Higher education, Scholarly communication, Open research

Funding: Research Office of Curtin University This work was funded by the Research Office of Curtin University through a strategic grant, the Curtin University Faculty of Humanities, and the School of Media, Creative Arts and Social Inquiry. The funders had no role in study design, data collection and analysis, decision to publish, or preparation of the manuscript.

==============================
While the movement for open research has gained momentum in recent years, there remain concerns about the broader commitment to openness in knowledge production and dissemination. Increasingly, universities are under pressure to transform themselves to engage with the wider community and to be more inclusive. Open knowledge institutions (OKIs) provide a framework that encourages universities to act with the principles of openness at their centre; not only should universities embrace digital open access (OA), but also lead actions in cultivating diversity, equity, transparency and positive changes in society. This leads to questions of whether we can evaluate the progress of OKIs and what are potential indicators for OKIs. As an exploratory study, this article reports on the collection and analysis of a list of potential OKI indicators. Data for these indicators are gathered for 43 Australian universities. The indicators provide high-dimensional and complex signals about university performances. They show evidence of large disparities in characteristics such as Indigenous employment and gender equity, and a preference for repository-mediated OA across Australian universities. We demonstrate use of the OKI evaluation framework to categorise these indicators into three platforms of diversity, communication and coordination. The analysis provides new insights into the Australian open knowledge landscape and ways of mapping different paths of OKIs.

Introduction

Demands on universities are changing, and universities need to change in order to meet these demands. Increasingly universities are interrogated about their effectiveness, impact, and accountability (Hazelkorn, 2018; Mader, Scott & Razak, 2013; Rubens et al., 2017). For example, the public wants to know how the taxpayer’s money is used to drive positive changes in society and students want to be able to evaluate the cost-effectiveness of the qualifications they pursue (Woodall, Hiller & Resnick, 2014). While some progress in meeting such demands is evidenced in the literature, many universities and departments still fall behind through a landscape that is unevenly positioned to support change (Gagliardi & Wellman, 2015; Perkmann, King & Pavelin, 2011). This is partly driven by the lack of clearly curated proxies for decision making, together with resource limitations, making it difficult for universities to determine the best strategies to implement.

University rankings somewhat fill this gap by providing simple league tables of university performances. These rankings (and related metrics) have rapidly become dominant in influencing resource allocations, employment, student choices, management strategies, and beyond. They drive behavioural changes to various levels of the university ecosystem (Hazelkorn, 2008, 2009; Niles et al., 2020). However, the narrowly defined set of metrics used by these rankings are often low-dimensional and the information they provide are confined to very specific measures (Johnes, 2018; Selten et al., 2020). They are also often criticised for their methodological shortcomings and a number of unintended side effects (Kehm, 2014; Goglio, 2016).

More recently, there is a strong focus on making university research more “open”. This is currently spearheaded by initiatives such as Plan S, which demands funder-supported research publications to be made open access (OA). While such movements for open research have gained momentum, there remain concerns about the broader commitment to openness in knowledge production and dissemination. For example, most existing academic reward systems fail to encourage open practices such as transparency, openness and reproducibility (Nosek et al., 2015). There are some signals of change in how universities are evaluated, as evidenced by the Times Higher Education’s Impact Rankings that assess performance against the United Nations’ Sustainable Development Goals (SDGs), and the inclusion of OA and gender indicators in the CWTS Leiden Rankings. However, there is a need for a clearly structured framework that is able to capture a university’s multidimensional efforts for achieving open knowledge goals.

Montgomery et al. (2021) describes one such framework in terms of open knowledge institutions (OKIs). It advocates for universities to act with the principles of openness at their centre; not only should universities embrace digital OA, but also lead actions in cultivating diversity, equity, transparency and positive changes in society. OKIs can be defined more formally as in the following quote:“We define an Open Knowledge Institution as an institution that does more than simply support or mandate specific practices. An effective Open Knowledge Institution embodies core values that deliver the benefits of open science (research) culture and practice. It achieves this through providing an environment, platforms and culture that deliver and also hold in tension three key areas: communication, diversity and coordination. Cultural change at institutional level and in response to national or regional initiatives and mandates is fundamental to achieving openness.” – Montgomery et al. (2019).

We adopt the above broad definitions of OKIs and “open knowledge” to enable wider dialogue and engagement with the multidimensional and interconnected aspects contributing to processes of knowledge creation and dissemination. It is our aim for universities to engage deeply with issues concerning the progression of OKIs and a broad framework gives OKIs the ability to contribute to change in the ways that universities are positioned and understood (Montgomery et al., 2021). An evaluation framework for OKIs is also proposed in Montgomery et al. (2021), where potential indicators are categorised into three platforms of diversity, communication and coordination. This is combined with a theory of change that evolves through aspiration, action and outcome, as signals of progress from narratives and policies, through investment and resource allocation, on to the delivery of results.

Data and results reported in this article serve to demonstrate the use of the above evaluation framework as a tool for mapping the performances of OKIs. As an exploratory study, we collect data on a number of OKI indicators for 43 Australian universities (universities listed under Table A and Table B of the Higher Education Support Act 2003 https://www.legislation.gov.au/Series/C2004A01234 and Avondale University College). These include indicators related to OA, collaboration, output formats, physical and online accessibility, Indigenous employment, gender equity, policy and infrastructure, and annual reports. We find a number of national anomalies such as large disparities in diversity of employment (which is negatively correlated to physical accessibility), and preferences for repository-mediated OA. Using robust statistical methods, we demonstrate that the signals provided by these indicators can be broadly categorised into the three platforms of diversity, communication and coordination. However, these signals are high-dimensional (i.e., diverse) with complex correlation structures, which also coincides with the theory of change described above. In summary, this study is guided by the following research questions:To what extent can the indicators available about Australian universities be described by the three platforms of openness as proposed by Montgomery et al. (2021)?

How can we better understand the relationships between these indicators and use them to capture the complexity of OKIs?

How does having access to these multidimensional indicators change our view of what would be more unidimensional rankings provided by traditional university league tables?

The main contributions of this article can be described in several parts. First, we present a diverse dataset of OKI indicators that potentially shape novel insights as to how universities interact with their surrounding communities. The dataset is also valuable in its own right for expanding further research on OKIs. Second, we provide a detailed analysis of relationships across these indicators. This leads to new understanding of the interconnections across different university activities and aspects for which the indicators represent. It also gives new supporting evidence for the use of the OKI evaluation framework as a plausible tool for mapping OKIs. Third, the results offer an alternative and broader view of university performances, as opposed to conventional league tables that focus heavily on research outputs. Altogether, the article also provides an outlook to further explorations in challenges of collection, integration, analysis and interpretation of a diverse set of interconnected indicators for OKIs.

The rest of the article is structured as follows. “Materials & Methods” introduces the data and indicators, with discussions on potential signals that these indicators may reveal. A brief description of statistical methodologies is also provided. “Results” reports on the various data analyses. These begin with descriptive statistical analyses of the OKI indicators, followed by correlation analysis, principal component analysis (PCA), and cluster analysis, respectively. “Discussion” discusses and summarises the main findings, implications thereof, and limitations to the findings. The conclusion is given in “Conclusions”. Acknowledgements and a list of references follow, with a number of appendices included in the Supplementary information.

Materials & Methods

We have gathered information for a selected set of 26 OKI indicators, plus an indicator on university revenue, from a variety of data sources. While some of these were collected via semi-automated procedures, others were collected manually. We have selected potential OKI indicators through both consulting with the OKI framework of Montgomery et al. (2021) and extensive scoping and reviewing for data availability and accessibility. Publicly accessible data and data accessible to the higher education sector are given preference over closed sources. We acknowledge that this list of indicators is not complete. However, it provides an important outlook for the challenges in tracking and analysing data associated with OKIs. The dataset also provides a unique view of university performance at multiple levels and dimensions. Wherever possible, the data is focussed on the year 2017 (see Limitations section for exceptions). The list of indicators examined, their data sources and collection processes are described in Table 1.

Table 1 The list of indicators, their description and data sources.

Indicator name	Code	Description	Data source and collection	
Total OA	oa_total	The proportion of outputs from a university that are made freely available online, via either the publishers or repositories.	Our data on OA were obtained through the Curtin Open Knowledge Initiative (COKI) data infrastructure. For each university, its outputs were searched and collected from Web of Science, Scopus and Microsoft Academic. This is done through APIs for the first two databases and a data snapshot in the case of the latter. The set of outputs is then filtered down to those that have Crossref DOIs (as per metadata record). Subsequently, the OA status of each output is obtained from an Unpaywall database snapshot. The number of OA outputs for each university is counted and divided by the total number of output for that university. Readers are referred to Huang et al. (2020b) for further details on this data collection process and versions of data snapshots used.	
Gold OA	oa_gold	The university’s proportion of outputs made freely available online via publishers under any OA license.	Similar to the above, but with the number of OA outputs restricted to those that provide access via publishers and have OA licenses, as per Unpaywall metadata.	
Bronze OA	oa_bronze	The university’s proportion of outputs made freely available online via publishers but with no clearly defined OA license.	Similar to the above, but with the number of OA outputs restricted to those that are accessible via publishers’ websites but do not have OA licenses, as per Unpaywall metadata.	
Green OA	oa_green	The university’s proportion of outputs made freely available online via repositories, regardless of whether they are also available via the publishers.	Similar to the above, but with the number of OA outputs restricted to those that are accessible via repositories, as per Unpaywall metadata.	
Green only OA	oa_green_only	The university’s proportion of outputs made freely available online via repositories, but are not available via the publishers.	Similar to the above, but with the number of OA outputs restricted to those that are accessible via repositories but are not accessible via publishers’ websites, as per Unpaywall metadata.	
Output diversity	output_div	The coefficient of unalikeability (Kader & Perry, 2007) based on the types of outputs affiliated to the university. This measure ranges between 0 and 1, where a higher number is indicative of more diverse output types.	These output types are identified using Crossref’s “type” field. Again, this is obtained via the COKI data infrastructure. The output types include “journal_articles”, “book_sections”, “authored_books”, “edited_volumes”, “reports”, “datasets”, “proceedings_article”, and “other_outputs”. For each university, the number of outputs that fall in each of these categories is counted. The coefficient of unalikeability is then calculated across these category totals, which produces the final measure for this indicator.	
Total collaboration	collab_total	The university’s proportion of outputs co-authored with one or more other organisations.	Organisations are identified and matched against unique identifiers from GRID (https://www.grid.ac/). This data is curated through the COKI data infrastructure, as for the OA data. Institutional links are drawn through co-authorships in Microsoft Academic. This is in addition to institutional search results for 1207 universities globally (including the top 1,000 in the 2019 Times Higher Education World University Rankings) from the Web of Science and Scopus APIs. In other words, we count (for a given university) the number of outputs that are also attached to (i.e., co-authored by) at least one other organisation in our data. This is divided by the total outputs to obtain a proportional measure.	
Collaboration with Australian universities	collab_aus	The university’s proportion of outputs co-authored with one or more other universities from the list of 43 Australian universities.	Similar to the above, except here we count the university’s outputs that are also attached to at least one other Australian university in our study set.	
Collaboration beyond Australian universities	collab_other	The university’s proportion of outputs co-authored with one or more organisations not from the list of 43 Australian universities.	Similar to the above, except here we count the university’s outputs that are also attached to at least one other organisation not in our study set of Australian universities.	
Industry collaboration	collab_ind	The university’s proportion of outputs co-authored with one or more industry partners.	This is obtained directly from the 2019 CWTS Leiden Ranking’s indicator for industry collaboration (https://www.leidenranking.com/ranking/2019/list). This indicator covers data ranging from 2014 to 2017.	
Total events	event_total	The university’s proportion of outputs with at least one Crossref event.	This is determined by counting the number of outputs with existing events in the Crossref Events Data for each of the universities. This is normalised by the university’s total number of outputs. Data is collected and curated via the COKI data infrastructure. An event is an instance for which the output is referred to online through various venues tracked by the various Crossref agents. These venues include sites such as Twitter, Reddit, Wikipedia, Wordpress.com, Newsfeed, etc.
See https://www.crossref.org/services/event-data/ for a list of data sources for events. Also see Appendix A for the justification on the calculation of this indicator.	
Walk score	walk_score	This is an index of efficiency of the physical location of the university.	This is manually obtained from http://www.walkscore.com, by searching for the university’s main campus location. This is included as a proxy for physical accessibility of the university.	
Website score	web_score	This is a score assigned to the university’s website based on the W3C Web Content Accessibility Guidelines (WCAG) 2.0 Level A and AA requirements.	This is obtained from the Functional Accessibility Evaluator 2.0 at https://fae.disability.illinois.edu/ on the status of the university’s website at the last available day of 2017 through the Internet Archive (https://archive.org/web/). This is designated to indicate the university’s effort to make their website more accessible.	
Indigenous staff	indigenous	The university’s proportion of indigenous staff (out of all staff).	Data obtained from the Australian Government’s Department of Education, Skills and Employment website at https://docs.education.gov.au/node/46146	
Women above rank of senior lecturer	women_above_sl	The university’s proportion of women, out of all academic positions above senior lecturer level.	Staff gender diversity data 2001–2018 downloaded as excel file from “uCube” http://highereducationstatistics.education.gov.au/Default.aspx using measures Staff Count, Current duties classification, Gender, Year, Institution. Staff count includes full-time and fractional full-time staff only.	
Women at rank of senior lecturer	women_sl	The university’s proportion of women, out of all academic positions at senior lecturer level.	As above.	
Women at rank of lecturer	women_l	The university’s proportion of women, out of all academic positions at lecturer level.	As above.	
Women below rank of lecturer	women_below_l	The university’s proportion of women, out of all academic positions below lecturer level.	As above.	
Women in academic roles	women_acad	The university’s proportion of women, out of all academic positions.	As above.	
Women in non-academic roles	women_non_acad	The university’s proportion of women, out of all non-academic roles.	As above.	
Policies on library access	policy_lib	The university’s score for library access policies.	This information is obtained by using Python scripts to search and checking manually on the university website for various characteristics. See Appendix B for more information.	
Policies on OA	policy_oa	The university’s score for policies and support for OA publications and data.	As above.	
Policies on diversity	policy_div	The university’s score for policies in equity and diversity.	As above.	
Annual report diversity score	ann_rep_diversity	The proportion of phrases in the university’s annual report that relate to diversity.	Annual reports (in PDF format) are collected from the university websites. Subsequently, a Python script is used to analyse word counts and number of occurrences of key phrases. See Appendix C for more detail.	
Annual report communication score	ann_rep_comm	The proportion of phrases in the university’s annual report that relate to communication.	As above.	
Annual report coordination score	ann_rep_coord	The proportion of phrases in the university’s annual report that relate to coordination.	As above.	
Total revenue	total_rev	The university’s total revenue, recorded in thousands of Australian dollars.	Data collected manually from the Department of Education, Skills and Employment (https://www.education.gov.au/2008-2017-finance-publications-and-tables).	

The inclusion of a number of OA indicators is intended to capture an OKI’s level of commitment on the different types of OA provision to their research outputs. These OA type categorisations are based mainly on whether an output is accessible via the publisher, repositories, or both, and whether it is assigned any OA license. The format of publication (e.g., journal articles, book chapters, conference proceedings, etc.) is often associated with disciplinary practices and represents diverse ways in which the university engages with its surrounding community. Similarly, the different collaboration indicators signal the demographic and geographic reach of the university’s research networks.

A less traditional indicator is the proportion of outputs with Crossref events (defined as an instance of mention or reference of a research output recorded over the web via the Crossref Event Data: https://www.crossref.org/services/event-data/. This can come from a number of sources such as Twitter, Wikipedia, Newsfeed, etc.). In essence, it is the proportion of research outputs visible online beyond traditional scholarly venues (see Appendix A for a justification on the way in which this indicator is calculated). It includes events such as social media mentions, citations in patents, and references in Wikipedia. This indicator is positioned as a signal of the university’s practices for online engagement and visibility, and its efforts in disseminating knowledge beyond the scholarly landscape (Sugimoto et al., 2017; Deeken, Mukhopadhyay & Jiang, 2020).

The “Walk score” and “Website score” are intended to give some indication of a university’s level of accessibility both physically (the former) and online (the latter). The “Walk score” takes into account walking distances and routes to nearby amenities, and pedestrian friendliness (see https://www.walkscore.com/methodology.shtml). Universities may not have direct and complete control over all of these factors, but many factors can be supported by universities. The factors that influence transportation modes include the availability of bus service routes, nearby amenities, on campus infrastructure, recreational buildings and population density (Sun, Oreskovic & Lin, 2014). As such, universities are well positioned to provide the support needed to reshape the transportation (and hence accessibility) patterns of its surrounding communities (Balsas, 2003). The “Walk score” also serves as a potential indication of university locations.

The “Website score” for each university’s main webpage is obtained via the Functional Accessibility Evaluator (https://fae.disability.illinois.edu/). This tool evaluates websites based on the W3C Web Content Accessibility Guidelines 2.0 Level A and AA requirements. In essence this measures how closely a website satisfies a list of recommendations to make its content more accessible to people with disabilities and more usable by individuals with challenging abilities due to aging. Meeting the W3C guidelines also makes web content more usable to general users. “Website score” is intended to be an indication of the university’s level of effort for making their outward facing content more accessible by diverse groups of online users.

We also include a set of measures for staff demographic diversity. In particular, we include the proportion of “Indigenous staff”, and proportions of women within various categories of university positions (as per Department of Education, Skills and Employment: https://www.education.gov.au/higher-education-statistics). These are signals of the university’s efforts in being inclusive in its knowledge production. Wilson et al. (2020) provides a detailed description of the challenges in collecting and interpreting such data at both national and international scales. We decided that it is useful to differentiate between academic and non-academic positions as they have different employment and evaluative criteria. Indeed, Larkins (2018) found a large disparity between the proportion of women in academic roles and the parallel figure in non-academic positions.

Lastly, we have a number of indicators related to the university’s policies and infrastructure surrounding a number of characteristics. These are intended to signal the university’s efforts in coordinating various facets of an OKI internally and across external communities. The first three indicators relate to policies on library access, OA, and diversity, respectively. In particular, a university’s score for each of these policy indicators is related to a predefined set of characteristics surrounding policy statements, regulations and provision for support. Further details of these indicators are provided in Appendix B and also see Wilson et al. (2019a; 2019b) for analyses of such indicators over international sample sets. We have also analysed each university’s annual report for keywords and key phrases surrounding diversity, communication and coordination. The indicators “Annual report diversity score”, “Annual report communication score” and “Annual report coordination score” are constructed as the number of times keywords or key phrases (from the predefined corresponding lists) appear in the document, divided by the total number of words in the document. It is worth mentioning that these indicators provide signals at varying degrees that are not necessarily direct signals of change or progress. However, they may provide signals of intent and proxies for institutional support to change (Montgomery et al., 2021).

We supplement the list of OKI indicators above with the “Total revenue” received by each university. This is aimed at serving as a benchmark for size and prestige. At the same time, we are interested in how total revenue may affect the OKI indicators.

The unit of measurement used for each indicator is carefully considered. The OA indicators, the collaboration indicators and “Total events” are represented as proportions of the institution’s total output to normalise against the size differences across universities. Similarly, “Indigenous staff”, the gender indicators and indicators on annual reports are reported in terms of the relevant proportions. On the other hand, “Output diversity” is calculated as a diversity measure dependent on the level of variability across output types. “Walk score”, “Website score” and “Total revenue” are left in their original units of measurement as per each data source. The policy indicators are the results of extensive manual document analysis and are presented as scores out of a number of characteristics relevant to each indicator (see Table 1 and Appendix B).

We emphasise that each indicator represents a composite piece of information that may provide various levels of signal (i.e., aspiration, action and outcome). This is in addition to complex relational structures across a number of indicators. As such, these indicators should often be considered together as a whole set, rather than studied individually, which could potentially miss a great portion of the full picture.

It should also be noted that our data does contain missing values. In particular, there are missing values in “Industry collaboration”, “Website score”, each of the gender indicators, each of the annual report indicators and “Total revenue”. Our subsequent analysis takes these missing values into account where applicable. These are differentiated from true zeros, which also exist in the data.

We utilise several statistical techniques to analyse this set of data. First, we focus on statistical descriptions that allow better comparisons across the different indicators. This is followed by an in-depth analysis of Spearman’s rank correlation to assess potential monotonic relationships between pairs of indicators. Robust PCA approaches are used to explore how the total variance across the OKI indicators can be best described by a smaller set of orthogonal principal components (PCs), and how the indicators relate to these PCs. Finally, cluster analysis performed on the data reveals clusters of universities, using the OKI indicators and resulting PCs as clustering criteria.

The codes and the relevant curated data for analysis are made available through Zenodo at https://doi.org/10.5281/zenodo.4040402. However, the raw data related to publications are not shared to respect the terms of service of the data sources. Links to codes and data for the policy analysis and the annual report analysis are also provided under the respective appendices in the Supplementary file. No ethics clearance was required for this study because statistical data are de-identified and publicly available.

Results

Descriptive data analysis

Our analysis starts with some individual descriptive statistics, and seeks insight into patterns emerging from each indicator and comparisons thereof. A number of summary statistics are recorded in Table D1 of Appendix D for reference. Here we focus on several statistical descriptions that are more comparative across the different measurements. Figure 1 below presents each indicator’s level of skewness, level of kurtosis, and its p-value resulting from the Shapiro–Wilk normality test. Many of the indicators exhibit substantial skewness (deviation from 0), and are leptokurtic (kurtosis value greater than the normal distribution, i.e., 3). These characteristics are consistent with the low p-values obtained for the corresponding Shapiro-Wilk test, indicating that many of the indicators are highly unlikely to be normal in distribution.

Figure 1 (A) Skewness, (B) kurtosis and (C) Shapiro–Wilk normality test p-value for indicators.

The proportion of “Indigenous staff” displays the highest levels in both skewness and kurtosis. It is positively skewed due to two universities having 16.13% and 4.4% Indigenous staff, respectively, compared to all other universities sitting in the 0% to 2.4% range. Other indicators with highest levels of skewness and kurtosis include “Gold OA”, “Collaboration beyond Australian universities ”, “Women above rank of senior lecturer”, and “Women in non-academic roles”. For each of these indicators, the data contains extreme observations (these are not outliers due to error and cannot be simply dropped). These extreme data points may have significant influences on the measures presented in Fig. 1.

For an alternative view of each indicator’s distribution, we construct their respective histograms. We use the Freedman-Diaconis rule for binwidths (hence, the corresponding number of bins) which is less sensitive to extreme observation than the standard Sturges’ formula for number of bins. This is applied to all indicators other than “Policies on library access”, “Policies on OA” and “Policies on diversity”, where the values of the original observations (i.e., scores) are used as bins. These histograms are displayed in Fig. 2.

Figure 2 Histograms for (A) oa_total, (B) oa_gold, (C) oa_bronze, (D) oa_green, (E) oa_green_only, (F) output_div, (G) collab_total, (H) collab_aus, (I) collab_other, (J) collab_ind, (K) event_total, (L) walk_score, (M) web_score, (N) indigenous, (O) women_above_sl, (P) women_sl, (Q) women_l, (R) women_below_l, (S) women_acad, (T) women_non_acad, (U) policy_lib, (V) policy_oa, (W) policy_div, (X) ann_rep_diversity, (Y) ann_rep_comm, (Z) ann_rep_coord and (AA) total_rev, respectively.

As mentioned, many of the indicators exhibit extreme observations. However, even with these extreme observations ignored (see Figure D1 of Appendix D for corresponding boxplots of normalized observations), many indicators are still characterised by substantial skewness (i.e., asymmetry) and high levels of kurtosis (i.e., fat tails). The above results imply the use of robust statistical methods that cater for non-normality and extreme observations is necessary for analysing this set of data.

To better understand each indicator, it is worth discussing some of these extreme observations. For example, the single extremely large observation for “Gold OA” corresponds to a significantly smaller institution in output size where 2 out of 3 publications were made OA via the publisher. The same university is responsible for the largest observations in “Total collaboration”, “Collaboration with Australian universities”, “Indigenous staff”, “Women above rank of senior lecturer”, “Women at rank of senior lecturer”, and the lowest values for “Collaboration beyond Australian universities”, “Industry collaboration”. Similarly, three of the smallest universities have 0% for “Bronze OA” and two of the smallest universities also contribute to the two lowest values in “Total events”. However, some extreme observations also result from other larger universities. For example, a medium to large sized university resulted in the largest value for “Annual report diversity score”.

The existence of extreme observations, together with highly diverse measurement units across the indicators, can easily distort standard measures of spread, such as variance and coefficient of variation. Hence, we supplement our analysis by calculating the quartile coefficient of dispersion (QCD) for each indicator. QCD is a unitless measure of dispersion that is more robust against extreme observations. This allows us to make an overall comparison of the relative dispersion across the indicators, and also study the level of disparity within each indicator. The results are displayed in Fig. 3, together with confidence interval (Bonett’s confidence interval, as appropriate for the sample sizes here; Bonett, 2006; Altunkaynak & Gamgam, 2019) for the QCD value in each case.

Figure 3 Quartile coefficient of dispersion (with confidence interval) for indicators.

Immediately we observe that “Total revenue” and “Indigenous staff” have the highest values for QCD, an indication of their high inequalities across the universities. The large inequality of “Total revenue” is highly correlated with output size (i.e., 0.97 in Spearman’s coefficient of rank correlation). This is immediately followed by the high level of disparity in the proportion of Indigenous staff (even with the effects of the extreme observations minimised by QCD). “Annual report communication score” also has a high QCD value, signalling the large differences across university annual reports in terms of the proportions of phrases or keywords related to communication. The “Walk score” indicator has the next highest QCD, which is likely attributed to the large differences in physical accessibility due to locations of the universities.

The proportion of collaborative publications (“Total collaboration”) and the proportion of non-academic women staff (“Women in non-academic roles”) have the lowest QCD value (ignoring the trivial case of “Policy on diversity”). This is a result of high concentration of values around the central part of their respective distributions (around 74% and 66% respectively). Overall, there is a high degree of differences in disparity across the indicators.

Correlation analysis

In this section, we explore the potential relationships between the OKI indicators. We calculate the (zero-order) Spearman’s rank correlation coefficient between all pairs of indicators (missing values are left out and only pairwise-complete observations are included for each pairwise calculation). The overall result is summarised into a network plot presented in Fig. 4, using multidimensional clustering. In the Figure, indicators that are more highly correlated appear closer together, and are joined by edges with darker shades. Blue edges indicate positive correlations, while red edges represent the negative correlations.

Figure 4 A network plot of Spearman’s rank correlation between OKI indicators.

In Montgomery et al. (2021), an evaluation framework for OKIs is proposed by characterising indicators into three platforms: diversity, communication and coordination. This is combined with a theory of change that evolves through the three stages of aspiration, action and outcomes. It also noted that the indicators may become increasingly more difficult to characterise into the three platforms as we move through the three stages of change. Figure 4 represents a practical example of the above. In the bottom right, we see a cluster of indicators mainly related to diversity (i.e., gender and Indigenous proportions). The bottom left section of the Figure encompasses a group of indicators related to communication, such as OA and collaboration. And, perhaps more dispersed, is a set of indicators for coordination (e.g., policies) gathered at the top of Fig. 4. Note that “Total revenue” is not included in Fig. 4, as it is not directly considered as an OKI indicator. We will take a closer look at its correlations with the OKI indicators later. It should be noted that the proximities of the points in Fig. 4 are determined by multiple clustering. This means where a point lies is relative to its magnitudes of correlations with all other points. Hence, direct comparisons between correlations of different pairs of indicators must be made with caution.

A few indicators appear less clearly defined in terms of which platforms they relate to, such as “Total events” and “Bronze OA”. This does not necessarily imply that they are not correlated with other OKI indicators. Rather, they may be similarly correlated (in magnitude) to indicators from multiple platforms making them less distinctive for classification. For example, “Total events” has similar magnitudes of correlation with both the OA indicators and the diversity indicators (see Fig. 5 for examples).

Figure 5 Scatterplots between ranks in (A) “event_total” versus ranks in “oa_gold” and (B) “women_acad”, respectively.

Another interesting observation is that “Output diversity” is closely located to many of the OA indicators and the collaboration indicators. These correlations are in the negative direction throughout. Two examples are given in Fig. 6. We should note here that most of the bibliographic data systems (including ours) capture more accurate information on journal articles than other output types. Also, given most universities have journal articles as the primary output format, the “Output diversity” indicator is mainly influenced by the inclusion of other output types, such as book chapters, conference proceedings, datasets, etc. The other output formats are also more likely to be recorded as non-OA1 (noting that our data workflow only tracks objects assigned with Crossref DOIs and is dependent on whether Unpaywall is able to accurately track their accessibility options) and are affiliated to smaller numbers of authors. Hence, higher diversity in output types is associated with lower OA levels.

Figure 6 Scatterplots between ranks in (A) “output_div” versus ranks in “oa_gold” and (B) “collab_total”, respectively.

The indicators “Collaboration with Australian universities” and “Walk score” appear out of place at the first glance. They seem more highly correlated to the gender and Indigenous indicators, and less correlated to other indicators relating to collaboration and access. However, a deeper exploration reveals some interesting relations. The “Walk score” is negatively correlated with the diversity indicators, while “Collaboration with Australian universities” is correlated positively with the same indicators. Examples of these for “Women in academic roles” are given in Fig. 7. Note that “Walk score” can also be seen (in general) as a benchmark for the locations of universities. This implies universities in more regional areas (which also tend to have lower total revenue and are smaller in size) have proportionally more women and Indigenous staff. These universities also produce higher proportions of outputs co-authored with others in the list of 43 universities (hence a negative rank correlation between “Walk score” and “Collaboration with Australian universities”). In contrast, “Walk score” is positively correlated with the other collaboration indicators, albeit at relatively smaller magnitudes.

Figure 7 Scatterplots between ranks in “women_acad” versus ranks in (A) “walk_score” and (B) “collab_aus”, respectively.

We also note the generally low correlation levels between OA indicators and the gender indicators. In comparison, Olejniczak & Wilson (2020) found that (for a sample of US universities) there is a slight bias towards male authors in terms of OA publications, and this bias increases if job security and level of resources are taken into account. Our data present little to no evidence for this bias in the Australian context, although further study is required.

Pairs of indicators with the highest positive values of rank correlation are “Total OA” against “Green OA” (0.89), and “Women at rank of lecturer” against “Women in academic roles” (0.86). In contrast, pairs with the highest negative correlation coefficients are “Gold OA” against “Output diversity” (−0.59), “Output diversity” against “Collaboration beyond Australian universities” (−0.57), and “Indigenous staff” against “Walk score” (−0.58). We have already had discussions related to pairs with highest negative correlations earlier. So we will now focus on pairs with the highest positive correlations. Figure 8 displays the scatterplots of the two pairs with the highest positive correlations.

Figure 8 Scatterplots between ranks in (A) “oa_total” versus “oa_green” and (B) “women_l” versus “women_acad”.

Almost all universities (42 out of 43) in our study have a higher proportion of “Green OA” (i.e., repository-mediated OA) than “Gold OA” (i.e., publisher-mediated OA), which is not unexpected as most gold OA publications are available to be archived by repositories (albeit sometimes in the form of earlier manuscript versions). For example, Robinson-Garcia, Costas & van Leeuwen (2020) showed that the scale of which “Gold OA” outputs are also archived by repositories to be as high as 81%. Green OA is also seen as the more cost-effective route compared to gold OA due to the high cost of article processing charges (APCs). In contrast, 34 universities have higher values in “Gold OA” than in “Green only OA”. Australian universities also have a low uptake on “Bronze OA”. The strong focus on “Green OA”, relative to other OA types, explains the high correlation between “Total OA” and “Green OA” (note for example the rank correlation coefficient is 0.55 between “Total OA” and “Gold OA” and is 0.89 between “Total OA” and “Green OA”).

The strong correlation between “Women at rank of lecturer” and “Women in academic roles” requires a deeper exploration. Figure 9 provides an overview of the proportion of women at various levels of academic positions within all 43 universities. Evidence shows there are higher proportions of women staff at lower academic positions consistently across all institutions. For example, 40 out of the 43 universities have a higher proportion of women in positions below lecturer level than proportion of women above senior lecturer level. There is also a higher proportion of women in non-academic roles, where 42 out of the 43 universities have a higher value for “Women in non-academic roles” than for “Women in academic roles”. It is also higher than all other gender proportions for almost all universities (as depicted in Fig. 9). Note that we also observed earlier (Fig. 3) that “Women in non-academic roles” has one of the lowest QCD values among all indicators. Together with a much larger number of women employed at the lecturer level in academic roles, the above factors contribute to the high correlation observed between “Women at rank of lecturer” and “Women in academic roles”.

Figure 9 Proportion of women at various levels of academic positions for all 43 Australian universities (anonymously labelled 1 to 43).

Lastly, we look at the potential influences of the university’s revenue on the OKI indicators. Figure 10 displays the rank correlation coefficient values between “Total revenue” and each of the other indicators. The three-way correlation between “Total revenue”, “Walk score” and “Indigenous staff” is consistent with our comment earlier regarding the sizes and locations of universities. That is, smaller, regional, and less wealthy universities are correlated with higher proportions of Indigenous staff. The negative correlation between “Total revenue” and all the gender indicators also displays a similar pattern. It is also worth noting that almost all universities in our data have existing policies on employment equity and diversity (as recorded by the “Policy on diversity” indicator). Evidently, some differences in outcomes exist across universities, but the levels of actioning (upon existing policies) remain difficult to quantify and require further exploration in the future.

Figure 10 Spearman’s rank correlation coefficient of “total_rev” versus each of the OKI indicators.

The correlation of “Total revenue” versus “Collaboration with Australian universities” and “Collaboration beyond Australian universities”, respectively, are high but in opposite directions. Consistent with our earlier discussion, universities with lower total revenue appear to have more proportions of output co-authored with other universities in our list of 43 Australian universities. In contrast, the more wealthy universities seem to have higher collaboration proportions outside these 43 universities, including international universities and research organisations. The size of “Total revenue” seems to have little correlation with “Industry collaboration”, though this indicator is derived from an external source directly and should be interpreted with caution.

The generally low correlations between “Total revenue” and the OA indicators (except for “Bronze OA”) is an interesting outcome. They are a potential indication that higher revenues at Australian universities do not necessarily translate to higher proportions of OA outputs (which one might expect in relation to the provision of OA publishing funds, the cost of APCs and costs associated to managing an institutional repository). The findings do however conform to the fact that only 3 universities in our data have signalled a provision of funding for OA publishing. This, and the fact that all but one university have an OA repository, may explain the slight increase in correlation against “Green only OA”. We should also note that the OA indicators seem to have only low to moderate correlations with “Policies on OA”. The moderate correlation between “Bronze OA” and “Total revenue” poses an interesting case, where we find more wealthy universities to have higher proportions of Bronze OA publications. However, these proportions remain generally low for all universities in the study.

Principal component analysis

In this section, we apply PCA on the OKI indicators. This is aimed at providing insight into how information is attributed across the different indicators, and how these indicators relate to a few principal components (PCs). Due to the existence of extreme observations, missing data and vastly different measurement scales, with no existing knowledge of a plausible re-scaling method, we propose a two stage process for PCA.

Firstly, being constrained by the size of data, we propose imputing the missing values using an iterative PCA algorithm (implemented by using the R package missMDA, with the function estim_ncpPCA used to estimate the number of PCs to be used for imputation and the iterative PCA process excuted by the imputePCA function). This procedure uses the mean of each indicator as initial values for the missing data. Subsequently, a standard PCA is applied and a selected number of PCs are used to re-estimate the missing values. The process is repeated iteratively until the imputed values converge.

Once the imputed data is obtained, we proceed with robust PCA procedures that cater for extreme observations. Common approaches for this purpose include the use of robust covariance (or correlation) matrices and projection pursuit. An example of the former is the use of the Spearman’s rank correlation matrix (equivalent to applying standard PCA to ranks within each indicator) in the classical PCA procedure. For the latter, one can use the ROBPCA procedure which combines robust covariance matrix estimation with projection pursuit (Hubert, Rousseeuw & Branden, 2005). Given that the results are very similar, we report only according to the first method in this section. Supplementary results and results for the second PCA method are provided in Appendix E.

Figure 11 shows the percentages of variance explained by each of the PCs and the corresponding eigenvalues derived from the Spearman-PCA approach. For the left diagram, the red line records the cumulative percentages of variance. While the first two PCs display the highest proportion of variance explained, their combined coverage is merely above 40%. It is observed that we need at least 8 PCs to attain a variance coverage of approximately 80% for the Spearman PCA. Analogous results are observed using the Kaiser criterion (red line in the diagram on the right) on eigenvalues to determine the number of PCs to retain.

Figure 11 Percentage of variance explained by PCs (A) and Scree plot of eigenvalues with Kaiser Criterion (B), for Spearman PCA.

The low proportions of variance explained by the individual PCs and the high number of PCs needed to attain a significant coverage of the cumulative variance indicate that the set of OKI indicators provides diverse information where the overall variance is spread across multiple directions. However, keeping a high number of PCs also makes the interpretations of these PCs more difficult, as the later PCs show less distinctive groupings of loadings by the original indicators (see Tables E1 for loadings on the first 8). A common approach is to rotate the PCs according to a select first few PCs based on informed judgement. In this case, the first two PCs display resemblance of the OKI evaluation framework described earlier; with one of the first two PCs correlated largely to diversity indicators while the other with the communication indicators. To a lesser extent, there may be a third PC that relates to coordination. As a graphical illustration of this, the correlation circle plot against the first two PCs is provided in Fig. 12.

Figure 12 Correlation circle against first 2 PCs using Spearman PCA.

In the correlation circle, all 26 OKI indicators are projected onto the first two PCs. Angles between arrowed lines represent correlations between indicators in this plane (with 90 degrees indicating zero correlation and 180 degrees indicating perfect negative correlation). Lengths of the arrowed lines are indicative of how well they are represented in this two-dimensional space (or their levels of contribution to these PCs). In Fig. 12, many of the diversity indicators are pointing to the right along the first PC, while many of the communication indicators are pointing in the direction of the second PC. Many of the correlations represented in this correlation circle are also consistent with observations made in Fig. 4 earlier. Most of the coordination indicators have shorter arrowed lines, indicating they are not well-represented in this space, and are likely to be more representative by other PCs.

The observations made above lead to a focus on the first three PCs (a third PC included to try capture variances in the coordination indicators). Table E3 in Appendix E lists the loadings by each of the OKI indicators after rotation (performed using the varimax function in R) against the first three PCs. For a simplistic overview that aligns with the OKI evaluation framework, we summarise the results into what proportions of each PC’s variance (calculated by sums of squares of standardised loadings of the selected indicators) are loaded by each of the three groups of indicators (diversity indicators: “Indigenous staff”, all gender indicators, “Collaboration with Australian universities” and “Walk score”; communication indicators: “Total OA”, “Gold OA”, “Green OA”, “Green only OA”, “Output diversity”, “Total collaboration” and “Collab beyond Australian universities”; coordination indicators: “Website score” and policy and annual report indicators. Others: “Total events”, “Bronze OA”, “Industry collaboration”). The groupings are decided depending on observations made in Fig. 4, and in conjunction to the Spearman’s rank correlation matrix and PCA loadings. Three of the indicators, “Total events”, “Bronze OA” and “Industry collaboration”, with non-distinctive grouping, are listed separately. These are presented in Table 2.

Table 2 Proportions of rotated PCs’ variances loaded by groups of OKI indicators.

	Spearman PCA	
Platforms	PC1	PC2	PC3	
Diversity	78.2%	2.7%	15.5%	
Communication	10.1%	84.3%	7.9%	
Coordination	3.6%	3.4%	61.6%	
“event_total”	2.3%	4.6%	13.6%	
“oa_bronze”	2.9%	2.9%	1.1%	
“collab_ind”	2.9%	2.1%	0.3%	

The results in Table 2 re-affirms our discussion earlier regarding which groups of variables provide the most loadings on the first two PCs. In addition, we observe the significant level of loading of the coordination indicators on a third PC. The indicator “Total events” loads on the coordination PC. On the other hand, “Industry collaboration” and “Bronze OA” seem to have little influence on the first 3 PCs.

Cluster analysis

In this section we are interested in exploring whether there are specific groupings of universities that can be defined by the OKI indicators. This is important in performing likewise comparisons and for identifying different paths of OKIs. As an immediate follow up from the PCA analyses, we are able to construct individual component plots of universities mapped onto any pair of PCs. In these plots, universities having similar profiles (or scores) in terms of the selected PCs will be displayed closer together. Figure 13 displays the individual component plot for the first two PCs from the Spearman PCA, with universities colour-coded by state or territory (as per main campus location), i.e., Australian Capital Territory (ACT), Multi-state (AU), New South Wales (NSW), Northern Territory (NT), Queensland (QLD), South Australia (SA), Tasmania (TAS), Victoria (VIC) and Western Australia (WA). No immediate pattern arises in terms of universities from a common state or territory as each group seems to be randomly scattered.

Figure 13 Individual component plot for first two PCs from Spearman PCA, with universities coloured by state.

Alternatively, we assign a colour to each university by their affiliation to existing Australian university networks (i.e., Australian Technology Network (ATN), Group of Eight (Go8), Innovative Research Universities (IRU), Regional Universities Network (RUN), and with universities unaffiliated to any network grouping labelled as “None”) in the same plot. This is presented in Fig. 14. The standout group is the Go8 where all 8 member universities lie towards the top-left of the plot. This is an indication that these universities are quite similar in terms of their performance in the first PC (diversity) and the second PC (communication). Their overall performance leans toward the top half in communication, but tends toward the opposite direction for diversity.

Figure 14 Individual component plot for first two PCs from Spearman PCA, with universities coloured by university network.

We next consider cluster analysis of universities using the full set of OKI indicators and the respective ranks. Columns are standardized (through subtracting by the column mean and dividing by the column’s mean absolute deviation) to cater for the differing units of measurement. Hierarchical clustering is implemented by using the Manhattan distance (only pairwise-complete observations are used) to construct the dissimilarity matrix between universities, and the complete-linkage criterion is used to select similar clusters (implemented using a number of functions and packages in R: the daisy function from cluster is used to calculate the Manhattan distance matrix; hclust is used for the cluster analysis and results are subsequently converted to a dendrogram object for plotting using the dendextend package). These selections are made based on their robustness against extreme values.

Dendrograms of the clustering results are presented in Figs. 15 and 16. The former is derived using ranks as input, while the latter uses the original observations. In both figures, the university labels are colour-coded by university network affiliations as before. Corresponding figures colour-coded by states and territories are given in Appendix E. The heights of the vertical lines indicate the order in which the clusters were joined, and that order is determined by how similar two universities are in terms of their performances (or ranks) in the OKI indicators. Universities joined with lines at lower heights are considered more similar to each other.

Figure 15 Dendrogram of hierarchical clustering of universities using ranks in OKI indicators, with universities coloured by university network.

Figure 16 Dendrogram of hierarchical clustering of universities using OKI indicators, with universities coloured by university network.

As before, the university networks reveal more synchronised groups in comparison to locations. The most prevalent case is that of the Go8 universities. In both figures, these universities appear to be closely clustered. This is a potential indication of the synergies across universities in common affiliated networks.

A further interesting observation is related to those universities that largely remain in singular or very small clusters. These extreme cases are less obvious in Fig. 15 given the use of ranks removes the size effects. However, both figures consistently show that many of the last few universities to be added to clusters are small, private or specialist universities. Intuitively this makes sense given that such universities may have less resources, missions that deviate from traditional universities, and practices that need to be aligned to these.

Discussion

Main findings and implications

In the previous section, we examined patterns and potential relationships across a number of OKI indicators for 43 Australian universities. We also explored ways in which information provided by these indicators can be summarised into a small number of orthogonal variables (PCs) in combination with the evaluation framework proposed by Montgomery et al. (2021). Universities were also clustered by using their corresponding data, ranks, and the corresponding PCs to reveal overall similarities across universities based on the OKI indicators. Based on these results, we reflect below on the research questions set out in the “Introduction” section.

First, we have found most of the OKI indicators to be clustered around the three platforms (i.e., communication, diversity and coordination) as described by Montgomery et al. (2021). This is signalled through both the correlation analysis and the PCA results. The network plot shows evidence of the three groupings through multiple clustering of the Spearman’s rank correlation coefficients. A few indicators (e.g., “Total events”, “Bronze OA” and “Industry collaboration”) seem less clearly positioned as to which platform they belong to. These are indicators with similar correlation levels against two or more of the three groups. However, if we consider these as signals of action or outcome (rather than aspiration), then their indistinctive positioning conforms to the theory of change within the OKI evaluation framework (Montgomery et al., 2021). For example, “Total events” is likely to be an outcome of mixtures between policy actioning (evaluation of “impact”) and communication (online accessibility to research), but also diversity (disciplinary practices), making its classification into the three platforms less trivial.

Through initial PCA results, we found that a higher number of PCs is needed to capture a reasonable proportion of the total variance. However, mappings of the OKI indicators onto the first few PCs show many diversity indicators aligning with the first PC and the communication indicators aligning with the second PC. Guided by this, the network plot and the OKI evaluation framework, we further rotated the PCs against the first three PCs. Examining the loadings onto the three rotated PCs reveals that diversity indicators make up most of the first rotated PC’s variance. Similarly, communication and coordination indicators make up most of the variances in the second and third rotated PCs respectively. These provide ways in which information underlying these indicators can be summarised, and demonstrate the use of the OKI evaluation framework as a potential tool for mapping performances of OKIs.

The second research question pertains to the understanding of relationships between the OKI indicators and their use in capturing the complexity of OKIs. Statistical analysis shows that many indicators are characterised by skewness and extreme observations. Hence, robust methods are needed to examine relationships across these indicators. The results also reveal large disparities across universities in terms of their performances in these indicators.

Investigation of correlations across these indicators finds interesting trends. Some trends are consistent with existing literature, while others indicate a shift of spectrum in the Australian context. For example, our data shows that Australian universities have relatively more of their research outputs made accessible via the “Green OA” route than the “Gold OA route”. This focus on “Green OA” is in agreement with trends on a more global scale for 2017 (Piwowar et al., 2018; Montgomery et al., 2021). This is likely a result of policy implementation and resource allocation in the Australian higher education sector. In comparison, there are regions for which “Gold OA” dominates the OA publication landscape, such as Latin America, based on pro-active policies and the SciELO (Scientific Electronic Library Online) cooperative OA publishing model (Huang et al., 2020b).

We also find increases in gender inequality moving up in academic ranks and between non-academic and academic positions. These are consistent with existing literature that shows progress for gender diversity to remain at more junior and non-academic positions within universities (Winslow & Davis, 2016; Baker, 2016). This occurs despite numerous diversity, equity and inclusion policies and action plans in operation at universities (Khan et al., 2019; Marini & Meschitti, 2018; Subbaye & Vithal, 2017; Winslow & Davis, 2016; Baker, 2016). While there has been progress on gender equity in the Australian context (Larkins, 2018), our findings suggest that the current level of diversity in academic positions is disparate across universities. In contrast, the proportion of “Women in non-academic roles” is quite consistent across universities (with the lowest QCD value out of all indicators). There also appear to be negative correlations between all diversity indicators (including “Indigenous”) and “Walk score” (which is a benchmark for location). These trends re-emphasise barriers to progress in hiring practices and achieving equity and parity (especially at traditionally high-ranking institutions). They indicate much work remains for advancing employment diversity in the Australian academic landscape.

Correlations of the OKI indicators against “Total revenue” also show several interesting patterns. Universities with higher “Total revenue” (also traditionally more prestigious) are correlated with higher proportions of “Collaboration beyond Australian universities”. The opposite is true for lower ranked universities who produce more proportions of “Collaboration with Australian universities”. This may be attributed to the fact that international collaboration enhances an institution’s reputation, impact, and ability to attract research and development investments, and research talents through both researchers and students (Australian Academy of Humanities, 2015; Glänzel, 2001; Glänzel & de Lange, 2002). These in turn influence the university’s position in international rankings. There is also a low correlation between “Total revenue” and “Gold OA”. In contrast, Siler et al. (2018) suggests gold OA publishing to be correlated to levels of funding and university ranks (although their finding is restricted to the field of Global Health research). This again conforms to Australia’s stronger focus on repository-mediated (Green) access to research outputs.

The above evidence suggests that analysis of these indicators in combination can support existing understandings of universities as OKIs and prompt new ways to interconnect these understandings. While each indicator is valuable in its own right, as a connected set of measurements, they can depict the complexity of OKIs. This offers novel insights into connections between these indicators and implications for universities’ progression as OKIs.

Access to these indicators provides alternative views on university performance. These multidimensional perspectives differ significantly from traditional university league tables that are more unidimensional. Indeed, our PCA results show that low levels of variance are explained by individual PCs before rotation and a high number of PCs is needed to explain a significant portion of the total variance. These imply that the information provided by these OKI indicators are high dimensional and complex. This is in contrast to findings related to popular university rankings where only one or two PCs are needed to capture most of the total variance in ranking measures (Dehon, McCathie & Verardi, 2010; Docampo, 2011; Johnes, 2018; Selten et al., 2020).

The complexity inherent in these indicators can be simplified through the use of the OKI evaluation framework. Together with cluster analysis, we are able to capture comparative performances of Australian universities on different groups of OKI indicators. There are both similarities and dissimilarities across universities, indicating their progress on a variety of paths as OKIs. Our results also suggest that classification of these universities as OKIs may be aligned with the Australian university network groups and levels of wealth and prestige. For example, universities in the Go8 are frontrunners in many of the communication indicators, but are relatively less successful in the diversity indicators. Smaller and less wealthy universities seem to illustrate higher performance in diversity but vary widely in their communication scores. In contrast, small, private and specialist universities portray performance patterns that are often unique compared to other universities.

Limitations and further research

While this study aims to be as comprehensive as possible in terms of both data collection and data analysis, a number of limitations need to be noted. Research outputs counted as part of this work are limited to those with existing DOIs that map with our Crossref data snapshot. Hence, a research output without a recorded Crossref DOI is left out. Affiliations between research outputs and universities come from three potential sources: Microsoft Academic, Web of Science, and Scopus. Each of these sources has their own limitations in accurately recording author affiliation information. Readers are referred to Huang et al. (2020a) for a report on such limitations. The bibliographic data sources are also dynamic (i.e., continuously changing), including potential backfilling. Our dataset focuses on the year 2017 with some exceptions. These are “Industry collaboration” and “Walk score”, where the former is reported by the CWTS Leiden Rankings for research outputs published over the period 2014 to 2017 and the latter represents scores shown at the time of access (2019) to the Walk Score website tool.

There exist missing values in the data that we have collected for the set of indicators. We have implemented and trialled various methods for the robust handling of these missing values. The results obtained are largely consistent across different methods. However, we cannot discard the possibility for the (unobserved) real data associated with these missing values to be vastly different and to potentially change the results significantly. Similarly, there are a number of extreme observations in the data, including those driven by sample sizes. While we have used robust methods to counter the potential size effects of these extremes, they nevertheless remain the largest or smallest values within the respective set of indicator observations.

We used extensive reviewing and manual work for the document analysis of university policies and statements for constructing the policy indicators. However, we cannot completely discard the possible subjectivities on our part of determining scores for these indicators. The annual report analysis is tested on two separate machines using different Python versions, and have yielded similar general results. However, we note that the process of transforming PDF documents to text files can depend on the operating system and versions of software used.

We have also subjectively chosen to use “Total revenue” as an indicator for university finance, size and prestige, but could have easily used other financial performance indicators (such as total expense, total grant income, fees and charges, employee benefits, etc.) from the Department of Education, Skills and Employment’s report. However, most of these indicators are highly correlated, e.g., the correlation between “Total revenue” and total expenses is more than 0.99, meaning they will likely lead to very similar results as reported in this article. Nonetheless, further explorations may be needed for less correlated financial indicators.

Lastly, our analysis is focused on the small sample of Australian universities for a specific year. Further research is needed to ascertain whether the main findings can be generalised to a larger region or over a longitudinal data set. The focus on a specific period also limits our ability to see potential time trends in these indicators and their relations. For example, it may be more informative to study “Walk score” over time to see the impact of universities on their surrounding environments. However, there remain challenges for collecting data over larger scales. Future research could also build on expanding the set of potential OKI indicators. For example, analysing the diversity of user groups that access the university website, library and publications may be useful. Further investigation is needed to examine the accessibility to such data.

Conclusions

In this study, we introduced a select number of OKI indicators that were collected by the Curtin Open Knowledge Initiative (COKI) project based at Curtin University. We explored several techniques for analysing these indicators that are robust towards catering for missing values, extreme observations and different measurement scales. The indicators provide high dimensional and complex signals for the progression of Australian universities as OKIs. The work also demonstrates the use of the OKI evaluation framework proposed by Montgomery et al. (2021) as a tool for mapping the performances of OKIs. This framework groups the OKI indicators into three platforms of diversity, communication and coordination, which are interpreted with a theory of change that evolves through aspiration, action and outcome. The value of considering a diverse set of indicators is that we provide new support for existing literature on university performance and novel insights to the Australian open knowledge landscape. This is achieved through connecting various aspects of the university community that may shift discussions about different paths for OKIs. It provides alternative and diverse views of the higher education sector that significantly differ from traditional university league tables. The study also provides outlooks for challenges in data collection and analysis that pave the way for expanding the research beyond the current set of OKI indicators and geographic reach.

Supplemental Information

Supplemental Information 1 Appendices to the main article.

Click here for additional data file.

The authors would like to thank the editors and reviewers for their valuable feedback that have helped to improve this article.

Additional Information and Declarations

Competing Interests

Author Contributions

Data Availability

1 For example, in the publication year 2017, Dimensions records the global proportion of OA articles to be around 50.7%, but only around 14.7% of edited books and monographs are OA. See https://app.dimensions.ai/discover/publication?or_facet_open_access_status_free=oa_all&and_facet_year=2017&or_facet_publication_type=book&or_facet_publication_type=monograph (accessed on 10 March 2021).

The authors declare that they have no competing interests.

Chun-Kai (Karl) Huang conceived and designed the experiments, performed the experiments, analyzed the data, prepared figures and/or tables, authored or reviewed drafts of the paper, and approved the final draft.

Katie Wilson conceived and designed the experiments, performed the experiments, analyzed the data, prepared figures and/or tables, authored or reviewed drafts of the paper, and approved the final draft.

Cameron Neylon conceived and designed the experiments, performed the experiments, analyzed the data, prepared figures and/or tables, authored or reviewed drafts of the paper, and approved the final draft.

Alkim Ozaygen conceived and designed the experiments, performed the experiments, analyzed the data, authored or reviewed drafts of the paper, and approved the final draft.

Lucy Montgomery conceived and designed the experiments, performed the experiments, authored or reviewed drafts of the paper, and approved the final draft.

Richard Hosking conceived and designed the experiments, performed the experiments, authored or reviewed drafts of the paper, and approved the final draft.

The following information was supplied regarding data availability:

Data and code are available at Zenodo: Huang, Chun-Kai, Wilson, Katie, Neylon, Cameron, Ozaygen, Alkim, Montgomery, Lucy, & Hosking, Richard. (2020, September 21). Codes and data for the analysis of open knowledge institution indicators: an Australian case study. Zenodo. DOI 10.5281/zenodo.4040403.

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
