# Peer review of "Mapping open knowledge institutions: an exploratory analysis of Australian universities"

_PeerJ, doi:10.7717/peerj.11391_

## Round 0.1 · original submission · Major Revisions

Two of the referees suggest major revisions and one minor so I am happy to invite a revision of the paper. You will need to provide a detailed response to the referees with your revised submission.

Reviewer 1 ·

Basic reporting

This article presents an exploratory analysis of 43 Australian universities using a unique dataset collected as part of the COKI Project (Curtin U). The article makes use of the Open Knowledge Institutions (OKI) framework (sharing some authors with this paper) to explore various dimensions of Australian Universities. In part, they discuss the way in which the available data can be used to analyze OKIs along three dimensions (all relevant to the concept of openness, according to the framework): diversity, communication, and coordination.

One of the article's strengths is the richness of the data analyzed, and the authors should be commended for making available detailed data used in their analyses. Making the data available and describing it are in and of themselves a contribution.

The article itself is clearly written, however, I fear that the main contribution to the article is somewhat lost in its presentation. More on this in the following section, but I would have liked to have seen something akin to research questions (RQs) laid out clearly at the end of the introduction. While I appreciate this is an exploratory study, the work is clearly guided by some RQs that are never identified as such.

With regards to organization, I would lastly note that the authors do not use the Results/Discussion/Conclusions in the traditional way. Their presentations of results is imbued with interpretation and commentary, while their discussions and conclusions are, as a result, left rather anemic. That is, the discussion is mostly found mixed in with the results and the discussion section is used to summarize the main results. While it is good to see a discussion section that walks us through these findings, I would encourage the authors to consider engage with the issues and literature that their article is addressing. That is, how do these main findings compare with what is already known? How do they shift the discussions taking place?

A note on presentation:
- It would make the paper a lot more readable if variable names were avoided entirely, and replaced with short common-sense names (eg. women_above_sl changed to "women above the rank of senior lecturer"). I understand this would add to the word count, but it would also make the paper much easier to read. This change, if adopted, should be seen in the text, tables, and figures. As an example of a paragraph I struggled to read as a result, see lines 227-235.

Experimental design

As hinted as above, I felt the paper was lacking clearly defined RQs. That said, I do think that the paper does explore some key issues that, had they been foregrounded as RQs from the outset, would have allowed the presentation of the richness of the data that the authors seemed to want to highlight. Three RQs are answered (to some extent), but never explicitly asked (naturally, these could be better formulated, but hopefully the following is sufficient convey my intent:
1. To what extent do the indicators available about Australian University cluster around the dimensions of openness proposed by Montgomery et al.? (ie. how "naturally" do the data fit into these categories)
2. How can these many indicators be used to capture the complexity of OKIs? (ie. How do these group together into Principal Components, and how many of those are necessary to capture the variance )
3. (perhaps the one least-well answered): How does having access to these multi-dimensional indicators change what would be a flat uni-dimensional ranking provided by league tables. (ie. how much do individual universities move in the rankings when more indicators are considered)

To be clear, answer to these RQs and the findings from them are presented in the paper, but the structure of the paper makes them emerge as secondary to the lengthy presentation of the underlying data, from which it is difficult, as a reader bombarded by the graphs and numbers, without guiding questions.

Notes methods:
- With the caveat that I am not an expert on statistical methods, I found the statistical comparisons to be appropriate and well-presented.
- I question the value of presenting both approaches to imputing the missing values for the purpose of the PCA. While I understand why the authors would feel compelled to try both, there appeared to be little different in the results of each, as least as far as they pertain to the main findings of the paper. As such, I suggest the authors indicate they did both, but present only one, since the conclusions remain unchanged.
- The use of Event Data requires further explanation, as it is not clear what this data includes, or to what extent it provides a useful signal (or of what).
- The use of the Walk Score is questionable, especially since this is an indicator, as the authors rightfully point out, of the university's geographical home and not of the characteristics of its openness—at least not a characteristic the university can do anything about. What is the point of including an indicator for which a university has no control? How does it change our understanding of how it functions, or how does it alter decision making? Do rural universities have satellite campuses? If so, are these accounted for in some way? This indicator raised more questions than answers for me.

Validity of the findings

Not sure if anything needs to be said here that isn't already captured in the above. I think the authors do draw valid conclusions from their data—namely that indicators for OKIs offer a more nuanced and multi-dimensional view of how institutions achieve openness.

Perhaps the only thing left to say is that, if the paper is to be revised along the lines suggested here, it would also require a more robust and targeted introduction/background section. The section, as written, glosses over the subject area without engaging with specific issues that need to be understood before the paper's findings can be interpreted (and that can then be re-engaged with in the discussion section).

Additional comments

There are a handful of statements that I flagged for myself in my initial reading. I point to them here without much detail, as they are not central to my critique of the paper (which is mostly structural):
- Why is Revenue the only additional indicator looked at, beyond the OKI indicators? What justifies including this, but not any other number of indicators that could be easily collected.
- The graph diagram of Correlations works really well to show clustering and strength of relationships. Going to use this one in the future, I'm sure!
- lines 301-308: this is discussion, not results (I had several of these, but I think this critique is clear enough above)
- line 238: don't think it's true most gold OA publications are archived. Is it?
- line 241: I don't think Bronze OA would be considered a "route" to OA
- line 317: distinction on what formats are included and which gets counted as OA by Unpaywall seems super important to emphasize.
- lines 377-392: those are a lot of questions that are never fully explored.
- lines 499-501: I wanted THIS discussed more. This is the key to the paper for me (my RQ2)
- lines 548-551: I also wanted THIS discussed more (my RQ3)

Reviewer 2 ·

Basic reporting

This article is well-written. There are a few typos and errors (e.g. 'facing resource limits on l61, 'summaries' instead of 'summarised' on l572, spelling error on x-axis of Figure 11) which doubtless will be picked up by the authors in their final polishing.

The reference list feels rather short for an article of this complexity, and several authors appear multiple times, including one of the co-authors of this paper. However, this is perhaps not surprising, as researchers in open research are still a small band.

The figures and tables are complete and raw data is provided.

Experimental design

As this article is an exploratory study, there is no research question included but the rationale for conducting the study is explained.

The methodology is well-described and the methods of analysis are well-justified. I suggest the authors should make clear who exactly selected the 26 indicators they chose to collect data on. This is not stated. It would be helpful to know by whom they were selected, on what basis, and what if any review and justification method was in place, and who acted as the reviewer.

There is no indication that the authors applied for ethics clearance. It's entirely possible it wasn't considered necessary but if so, this should be noted.

Validity of the findings

For me, the Discussion and Conclusion lacked content, compared to the richness of the data and analysis in the Results sections. A summary of the findings is not a discussion. As the aim of the research was to 'evaluate the progress of OKIs and ... potential indicators for OKIs', this section needs greater reflection on the quality, robustness and value of the selected indicators - as an exploratory analysis, an evaluation of the indicators would provide the underpinning for further study and avoid repetition by other researchers.

Additional comments

My major criticisms come from the introductory section of the article. The opening paragraph (ll56-62) lacks evidence. If these are your contentions, then that needs to be clearly stated but actually, there is ample evidence in the literature on public engagement with research (e.g the Public Attitudes to Science surveys in the UK) and higher education to substantiate the demands and needs identified in this paragraph.

Lines 73-76 - the discussion of open access, etc., conflate open science and open access, which are not the same thing. This can be clarified, I believe. It is also worth mentioning that Plan S has been adopted by funders beyond the sciences, so is a movement for open research, rather than open science. And as above, statements such as 'there remain concerns about the broader commitment to openness in knowledge production and dissemination' should be justified by evidence.

·

Basic reporting

Please see the attached review.

Experimental design

This was not an experiment.

Validity of the findings

Please see the attached review.

Additional comments

Please see the attached review.

---

## Round 0.2 · accepted · Accept

All three referees recommend accepting the paper.

Reviewer 1 ·

Basic reporting

see below

Experimental design

see below

Validity of the findings

see below

Additional comments

The authors have taken my recommendations to heart and have done a substantial revision in the organization of the paper. As a result, in my reading, the paper has been much improved. I hope the authors agree.

Important to point out that the core of the paper remains fundamentally the same—the data and methods are sound and the results have not changed—but the current version does a much better job at expressing the guiding questions and making those findings stand out in a way that was challenging in the first

I especially appreciated the new Discussion section which, in this version, adequately engages with other literature and compares and contrasts the study's findings with what can be found in the literature. It also much more clearly points out the ways in which the clustering of indicators can be useful for making sense of the multi-dimensional data.

I have no further suggestions for improvements and endorse the publication in its current form.

Reviewer 2 ·

Basic reporting

The revised article meets the required standards

Experimental design

The methodology is greatly improved by the inclusion of clear research questions.

Validity of the findings

The Discussion is much clearer and well-separated from the Results and is a much clearer evaluation of the indictaors.

Additional comments

The revisions have greatly improved the paper and I am content that you have addressed all my comments.

·

Basic reporting

See general comments

Experimental design

See general comments

Validity of the findings

See general comments

Additional comments

I think that the authors have made very thorough and thoughtful changes in response to the reviews, and I'm happy suggest that the article be accepted. I have a few minor points to raise, but (unless other reviewers raise objections necessitating another substantive revision) I think these could be flagged as discretionary revisions.

1. Introduction: Research question 1 asks about how the indicators "cluster", which is perhaps mildly confusing given that this research question is seemingly investigated using PCA. Cluster analysis is used in the article, but not for this question.

2. Materials and Methods: "The “Walk score” also serves as a potential indication of university locations and demographic diversity." It's not clear to me why the walk score might be considered as an indicator of demographic diversity(?)

3. It'd be worth checking the figure numbering before publication (e.g., "Figure 13 displays the individual component plot for the first two PCs from the Spearman PCA" - I think this should be Figure 14?)

4. Results: "Columns are standardized (through subtracting by the column mean and dividing by the column’s mean absolute deviation) to cater for the different measurement levels. " - I think this should probably be "differing units of measurement" rather than "different measurement levels" (standardising via a linear transformation wouldn't do much to address different levels of measurement).

5. Discussion: "For example, our data shows that Australian universities have a relatively stronger focus on “Green OA” than “Gold OA"". Do you mean that the universities published more articles as Green OA than as Gold? ("Stronger focus on" is a little ambiguous).

6. Discussion: "Our findings suggest that while there has been progress on gender equity in the Australian context (Larkins, 2018), the level of progress for academic positions is disparate across universities". I have no doubt whatsoever that the *conclusion* of this argument is correct, but given that it's a claim about change (progress) over time it is not really a claim that is supported by this study's findings per se. Perhaps rephrase to avoid the implication that the study measured progress over time.

7. Discussion: "At least in theory, this potentially implies Indigenous and female academics may need to relocate (perhaps to more regional universities) to seek potential promotion opportunities". I think some might read this sentence as prescriptive (i.e., a recommendation for what female and Indigenous academics should do) when you probably mean it descriptively (that Indigenous and female academics may sometimes be forced to relocate to regional universities due to differing hiring practices). And then the rest of the paragraph provides a reason to be skeptical about whether it is in fact hiring practices that are responsible for the observed difference anyway, at least for Indigenous scholars ("the levels of Indigenous employment may be driven by local and regional demographics"). So perhaps you might prefer not to include this sentence.